# GeoTimeCLIP: Unveiling the When and Where of Images

## Abstract

Timestamp prediction aims to accurately determine the date and hour at which an image was captured using only visual cues, with applications ranging from image retrieval and metadata correction to digital forensics. In outdoor scenes, this can be inferred from variables such as overall brightness, hue, and shadow positions for hourly estimations, as well as weather patterns or seasonal changes for determining the date. However, these factors vary greatly depending on geographical location, making the challenges of time-of-capture prediction closely related to geo-localization. To address this problem, we introduce GeoTimeCLIP, a novel method capable of simultaneously estimating both the capture time (i.e., hour and month) and geo-location (i.e., GPS coordinates) of an image using a retrieval approach. Our model employs an image encoder, a time encoder, and a location encoder, aligning the time and GPS embeddings with the image embeddings in a continuous high-dimensional feature space. Considering the cyclical nature of days and years, we propose an effective way to represent time using Random Fourier Features. To learn image-time embedding alignment, rather than applying a standard contrastive loss with hard positives and negatives, we propose a more effective metric learning-based objective, which provides soft targets by considering the time difference between samples over a toroidal manifold. We introduce new benchmarks for time prediction, where we show that our jointly optimized time-location-based method outperforms baselines optimized solely for time. We also evaluate our method on existing geo-localization protocols, demonstrating that our approach performs competitively with expert geo-localization methods. Our shared embedding space enables various downstream tasks, such as compositional retrieval and text-based retrieval.

## 1 Introduction

Estimating the capture time and geo-location of images is crucial for applications ranging from digital forensics to ecological studies and social media management. In digital forensics, accurate timestamps verify image authenticity and help detect manipulation, particularly when camera calibrations are suspect. This capability is essential for reconstructing events from timestamped images during accidents or natural disasters, providing critical information to first responders. Ecological studies benefit from time-ordered images to monitor changes in landscapes and wildlife, while precise timestamps in social media enhance content management and chronological sorting. Despite its importance, predicting time from images presents several challenges due to the intricate relationship between temporal cues and location-specific factors. Time of day and year manifest differently in images due to variables like scene brightness, shadows, weather, and seasonal changes, making it difficult to establish consistent patterns. The complexity of the task is further compounded as the visual appearance of specific hours vary substantially across different months and locations, influenced by the amount and relative exposure to sunlight. Additionally, the representation of months fluctuates across various latitudes, with regions near the equator experiencing relatively stable climate conditions year-round compared to regions at higher latitudes.

Most existing methods (Zhai et al., 2019; Salem et al., 2022; Padilha et al., 2022) rely heavily on GPS data for accurate time estimation, making the absence of such metadata a significant challenge. Conversely, state-of-the-art geo-localization models, such as PIGEON (Haas et al., 2024) and GeoCLIP (Vivanco Cepeda et al., 2024), can effectively predict locations at the country or continent

level. However, accurately predicting both time and location without relying on additional inputs remains an unaddressed challenge.

In this paper, we introduce GeoTimeCLIP, a retrieval-based approach for joint geo-localization and time prediction. We conceptualize time prediction as a retrieval problem, representing time as a month-hour pair. A schematic diagram of our framework is depicted in Figure 1. Building upon the CLIP-initialized visual model, our goal is to learn a shared embedding space where we can align visual (image), time, and location modalities. For time prediction, we first propose a time representation that considers the cyclical nature days and years. We then project these representation into a multi-scale, high-dimensional time embedding using random Fourier features (RFFs) (Tancik et al., 2020). Next, to learn the alignment between the time and image embeddings, we explore several possibilities. Existing contrastive learning methods, including CLIP (Radford et al., 2021) and SimCLR (Chen et al., 2020), use other batch instances as negative samples. While suitable for the image-location contrastive losses in CSP (Mai et al., 2023), GeoCLIP (Vivanco Cepeda et al., 2024) and SatCLIP (Klemmer et al., 2023), due to the significant variation of visual appearance with respect to geographical location, this approach does not work well for the image-time modality. As visual features vary smoothly over time, and adjacent hours or months often appear nearly identical, treating them as negatives severely impedes alignment. Instead of defining positive-negative pairs as in contrastive learning, we propose a novel *Temporal Metric Learning* approach, which encourages similarity between two instances based on their time difference. To build the target metric for our proposed loss, we use the toroidal distance between the times of each instance pair to consider the cyclic nature of time.

This approach enhances performance without the need for explicit assignment of positive and negative samples, providing a more effective and efficient solution for time prediction. By mapping the image, location, and time modalities into a unified feature space, our model gains the ability to perform compositional retrieval tasks. For instance, given a specific time and location, it can efficiently retrieve all corresponding images from a gallery that closely match the specified criteria.

Our main contributions are the following:

- A framework for joint time-of-capture prediction and geo-localization by aligning the image, time and location embeddings in a shared multimodal feature space using contrastive learning.
- First retrieval-based method for time-of-capture prediction where we propose the novel time representation in month-hour pairs, considering its cyclic nature.
- Novel Temporal Metric Learning based loss function for image-time alignment with soft targets, eliminating the need to assign positive and negative samples to the anchor. Since both hours and months are cyclic, we employ a cyclic toroidal distance instead of a regular L2 distance which results in improved performance.
- We propose new standard benchmarks for time prediction, demonstrating that our jointly optimized time-location method surpasses time-only optimized baselines and competes well with expert geo-localization methods. Our shared embedding space further facilitates downstream tasks like compositional and text-based retrieval.

## 2 RELATED WORKS

**Time-of-capture prediction:** Time-of-capture prediction is a relatively new problem that has only been directly addressed by a handful of prior works. Tsai et al. (2016) proposed a physically inspired method to infer the time of day by estimating the sun's position and camera orientation, but their approach requires sky visibility and additional metadata, such as GPS coordinates and access to an external image database. In a different line of research, Zhai et al. (2019) introduced a data-centric approach to learn geo-temporal image features. Their model uses an image, location, and time encoders to generate mid-level features, which are subsequently passed to a set of classifiers for predicting time and location as discrete classes. However, their evaluation shows that providing the location as an input is crucial for predicting the time of day with reasonable accuracy. Similarly, Salem et al. (2022) proposed a hierarchical model to predict the month, hour, and week of capture, but this method also assumes known geo-location, limiting its real-world applicability. In contrast, our model relies solely on images to generate accurate time predictions using a retrieval approach in

a continuous shared feature space, with resolution determined by a gallery of arbitrary size rather than discrete classes.

Other works have also explored the time-of-capture task indirectly. Li et al. (2017) presented an algorithm to verify image capture time and location by comparing the sun position, computed from the claimed time and location, with the actual sun position derived from shadow length and orientation. However, their approach assumes that latitude, time-of-day, and time-of-year are given, with only one potentially corrupted. Padilha et al. (2022) proposed a model for time-of-capture verification using a data-centric approach, involving four encoders for ground-level images, timestamps, geo-locations, and satellite images, fed into a binary classifier to predict time consistency. Similarly, Salem et al. (2020) proposed a model to generate a global-scale dynamic map of visual appearance by matching visual attributes across images annotated with timestamps and GPS coordinates. Both Padilha et al. (2022) and Salem et al. (2020) show qualitative results on time prediction, but are limited by their dependency on geo-location. In contrast, our method accurately estimates both GPS coordinates and timestamps using only images.

Several additional methods explore problems adjacent to time prediction. Jacobs et al. (2007) proposed an algorithm for geo-locating static cameras by comparing temporal principal components of yearly image sequences with those from a gallery of known locations. For shadow detection, Lalonde et al. (2012) used a multi-stage method to extract pixel-wise ground-shadow features and find edges using CRF optimization, while Wehrwein et al. (2015) used illumination ratios to label shadow points in a 3D reconstruction and compute dense shadow labels in pixel space. Another series of works estimate the sun position for various downstream applications, such as computing camera parameters from timelapses (Lalonde et al., 2010) and determining outdoor illumination conditions (Lalonde et al., 2012; Hold-Geoffroy et al., 2017) from single images. Adapting these methods for time prediction would require additional metadata, which might not be available during inference. For example, even with correct sun position prediction, day of the year, geo-location, and compass orientation are needed to accurately predict the hour.

Although the above works contribute to time prediction, they either require additional input metadata like GPS coordinates, or are not reliable in the absence of specific temporal cues. In contrast, our proposed GeoTimeCLIP aims to predict both time-of-day and month from a single image without relying on any additional metadata, making it more broadly applicable for real-time prediction tasks.

**Global geo-localization:** Geo-localization, the task of estimating the geographic coordinates of an image, has gained substantial popularity in recent years. Traditionally, geo-localization methods have adopted either a classification approach (Weyand et al., 2016; Seo et al., 2018; Vo et al., 2017; Muller-Budack et al., 2018; Pramanick et al., 2022) or an image retrieval approach (Regmi & Shah, 2019; Shi et al., 2019; 2020; Toker et al., 2021; Zhu et al., 2021; 2022). The classification approach divides the Earth into a fixed number of geo-cells, assigning the center coordinate of the selected class as the GPS prediction. However, this method can result in significant errors depending on the size of the geo-cells, even when the correct class is selected. In contrast, the image retrieval approach compares a query image to a gallery and retrieves the one with the highest similarity. GeoCLIP (Vivanco Cepeda et al., 2024) addresses the limitations of traditional approaches by framing global geo-localization as a GPS retrieval problem. It leverages the pretrained CLIP (Radford et al., 2021) ViT and employs contrastive learning to align image and location embeddings in a shared feature space. Other methods, such as PIGEON (Haas et al., 2024), use a hybrid strategy: first employing image classification to identify the top-$k$ geo-cells with the highest probability, followed by a secondary retrieval stage for refinement within and across geo-cells. However, PIGEON's dependence on additional metadata for training—such as administrative boundaries, climate, and traffic—poses a significant limitation. Finally, recent methods such as Img2Loc (Zhou et al., 2024) have begun leveraging multimodal large language models (MLLMs) and retrieval-augmented generation (RAG) to achieve competitive geo-localization performance without the need for dedicated training. However, this method's effectiveness is highly dependent on the underlying MLLM and results in substantial inference overhead.

**Geo-spatial dual-encoder methods:** The success of CLIP has inspired to leverage its architecture for geo-spatial tasks. SatCLIP (Klemmer et al., 2023) aligns satellite imagery and natural images in a shared feature space, enabling cross-modal retrieval and localization. In a similar fashion, Zavras et al. (2024) proposes a method for aligning complementary remote sensing modalities beyond RGB with the CLIP encoders. Other works from Mai et al. (2023) and Mac Aodha et al. (2019), employ

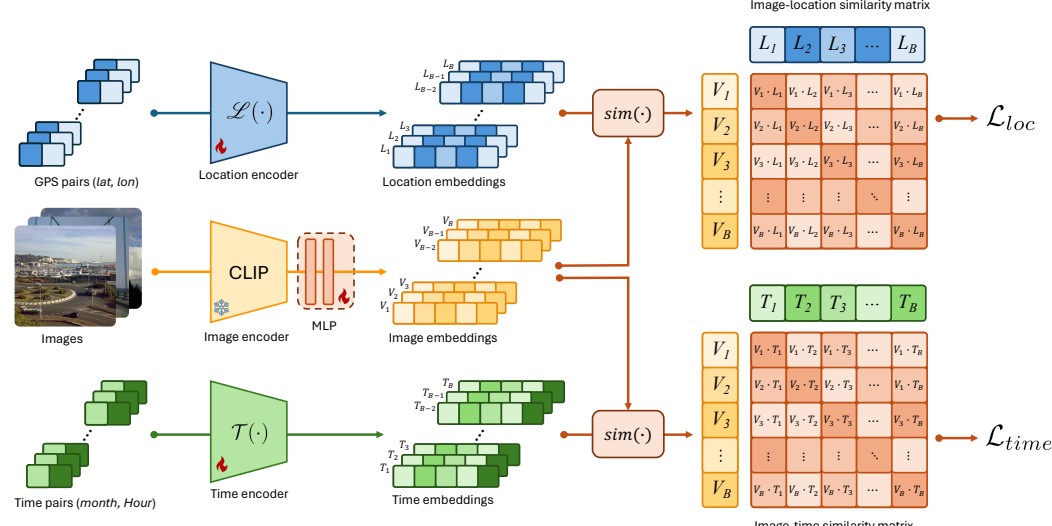

Figure 1: **Overview of GeoTimeCLIP**: GeoTimeCLIP uses an image encoder $\mathcal{V}(\cdot)$, location encoder $\mathcal{L}(\cdot)$ and time encoder $\mathcal{T}(\cdot)$ to generate a set of image $V_i$, location $L_i$ and time $T_i$ embeddings. Leveraging the CLIP Radford et al. (2021) pretrained ViT-L/14 as image encoder, we aim to align its image embedding to both location and time embedding. The image-location alignment is learned through regular CLIP like loss Vivanco Cepeda et al. (2024) and image-time alignment is learned through proposed Temporal Metric Learning.

a dual image-location encoder architecture to learn robust location representations from images. However, their ultimate goal is not to geo-locate images. Instead, they use the embeddings from the pretrained image encoder for downstream tasks like image classification. These methods demonstrate the effectiveness of dual-encoder architectures for geo-spatial problems, motivating our approach of using a triple encoder architecture for joint time and location prediction.

## 3 METHOD

Given a training dataset $S_{train} = \{(I_n, G_n, D_n)\}_{n=1}^{N}$ consisting of image $I_n$, GPS coordinates $G_n$ and timestamp $T_n$ triplets, our objective is to train a model that can simultaneously predict the location, time-of-day (ToD) and time-of-year (ToY) from unseen images. Our GeoTimeCLIP method consists of three encoders: Image Encoder ($\mathcal{V}$), Location Encoder ($\mathcal{L}$), and Time Encoder ($\mathcal{T}$) as shown in Figure. 1. Both geo-localization and time prediction are framed as a retrieval approach. Given a query image from the test gallery $I^Q \in S_{test}$, we compute an image embedding, $V^Q = \mathcal{V}(I^Q)$, using a pre-trained Vision Transformer. Similarly, given a gallery of latitude-longitude pairs, and a gallery of timestamps, we respectively compute galleries of location embeddings $L_k^Q = \mathcal{L}(G_k^Q)$ and time embeddings $T_k^Q = \mathcal{T}(D_k^Q)$. In order to predict the location and time, the image embedding is compared against both galleries. The GPS and capture-time with the highest similarity to the image are selected as predictions.

A fundamental prerequisite for retrieval is the alignment of image, location, and time modalities within a shared multimodal embedding space. To achieve this, our framework is optimized using two multimodal alignment objectives: (1) Image-Location alignment, and (2) Image-Time alignment. Furthermore, to capitalize on large-scale visual pretraining, we employ the pretrained CLIP ViT-L/14 as our image encoder, projecting it into our shared embedding space using a trainable 2-layer multi-layer perceptron (MLP).

### 3.1 IMAGE-LOCATION ALIGNMENT

We adopt GeoCLIP for the image-location modality alignment. Given a latitude-longitude pair $G_i$, it first uses Equal Earth Projection (EEP) to mitigate the distortion of the standard GPS coordinate

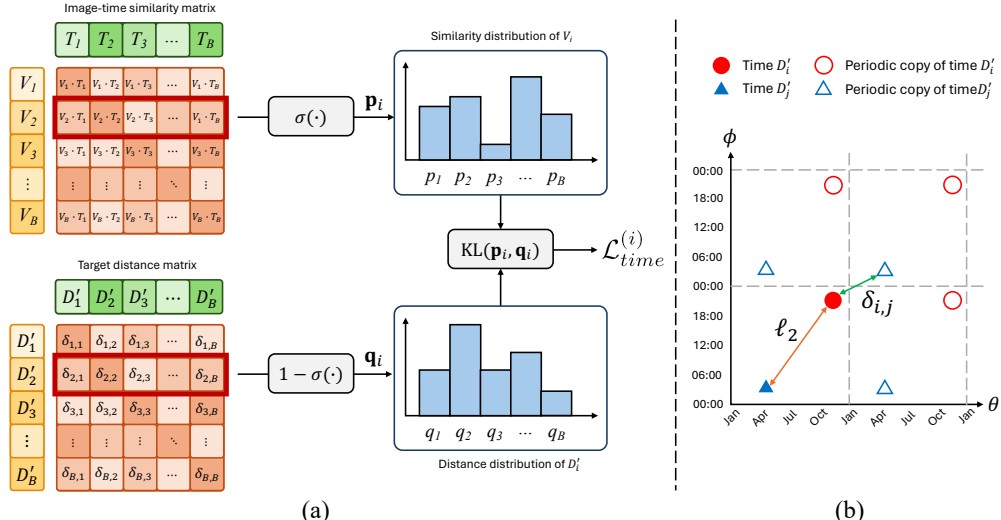

(a)             (b)

Figure 2: (a) **Proposed Temporal Metric Learning based loss** $\mathcal{L}_{time}$**:** We obtain the Image-similarity matrix by taking the cosine distance between the image and time embedding of the all instances of the batch. We obtain Target distance matrix by computing the toroidal distance of time of each instance pairs. As shown in the red highlighted box, we take the $i$th row of both matrices and normalize them using the softmax function $\sigma(\cdot)$ and $1 - \sigma(\cdot)$ respectively, resulting in two probability mass functions $\boldsymbol{p}_i$ and $\boldsymbol{q}_i$. The loss is then given by the KL-divergence between $\boldsymbol{p}_i$ and $\boldsymbol{q}_i$. (b) **Our proposed distance metric** $\delta_{i,j}$**:** Assume we have two normalized month-hour pairs $D_i' = (\theta_i, \phi_i)$ and $D_j' = (\theta_j, \phi_j)$. Since month and hours are periodic, they repeat themselves infinitely in both directions of the $\theta$-$\phi$ plane. Using the $\ell_2$ distance overestimates the real distance between the two times, but our proposed time distance $\delta_{i,j}$ is able to provide a correct estimate by considering the minimum distance between $D_i'$ and the periodic copies of $D_j'$.

system and provide a more accurate representation $G_i'$. Then, Random Fourier Features (RFF) are used to map the 2D representation into a rich high-dimensional representation at three scales ($M$) using projection matrices $\gamma(\cdot)$ with different frequencies $\sigma_i \in \{2^0, 2^4, 2^8\}$. Lastly, the RFFs are passed to a set of MLPs $f_i$ and added together, forming a single multi-scale feature vector. This can be mathematically expressed as the following equation:

$$L_i = \mathscr{L}(G_i) = \sum_{i=1}^{M} f_i(\gamma(EEP(G_i), \sigma_i)). \tag{1}$$

Next, to compute the image-location contrastive loss, we consider a set of $P$ augmented image $V_{ij}$ and location $L_{ij}$ embeddings ($j \in 1, \ldots, P$) and $S$ additional location embeddings $\tilde{L}_k$ ($k \in 1, \ldots, S$) stored in a continually updated dynamic queue of size $S$. For the batch with size $B$ and temperature $\tau$ the loss is given by:

$$\mathcal{L}_{loc} = -\sum_{i=1}^{N}\sum_{j=1}^{P} \log\left(\frac{\exp(V_{ij} \cdot L_{ij}/\tau)}{\sum_{k=1}^{B}\exp(V_{kj} \cdot L_{kj}/\tau) + \sum_{k=1}^{S}\exp(V_{kj} \cdot \tilde{L}_k/\tau)}\right). \tag{2}$$

### 3.2 IMAGE-TIME ALIGNMENT

**Time representation:** The capture time of an image is usually a Unix timestamp, an integer tracking the seconds (or milliseconds) elapsed since January 1, 1970. We discard the year information from the timestamp, as predicting the year is beyond the scope of this work, and instead focus on Time-of-Year (ToY, i.e., month) and Time-of-Day prediction (ToD, i.e., hour). Both ToY and ToD are cyclical, with periods of 12 months and 24 hours, respectively. To convert the Unix timestamp $U_i$ into a time representation that focuses on the months and hours, we transform it into a date tuple

$D_i = \texttt{unix2tuple}(U_i) = (m_i, d_i, H_i, M_i, S_i)$ with the month, day, hour, minute, and second. From this tuple, we then compute a new time representation comprised of the normalized cyclic month-hour pair $D_i' = (\theta_i, \phi_i)$ using

$$\theta_i = \frac{1}{12}\left((m_i - 1) + \frac{(d_i - 1)}{\mathcal{D}(m_i)}\right), \phi_i = \frac{1}{24}\left(H_i + \frac{M_i}{60} + \frac{S_i}{3600}\right), \tag{3}$$

where $\mathcal{D}(m_i)$ is the number of days in month $m_i$. We then convert a Unix timestamp to a normalized cyclic month-hour pair as $D_i' = \texttt{unix2cyclic}(U_i)$.

**Time encoding:** By representing time as a pair of continuous numbers, the problem of time prediction becomes similar in nature to geo-localization. Instead of retrieving the latitude-longitude pair with the highest similarity, we now want to retrieve a month-hour pair. Our time encoder ($\mathcal{T}$) follows the exact same architecture as the location encoder ($\mathcal{L}$). Similar to Eq. 1, time embedding is obtained from the proposed time representation from the following equation:

$$T_i = \mathcal{T}(U_i) = \sum_{i=1}^{M} f_i(\gamma(\texttt{unix2cyclic}(U_i), \sigma_i)). \tag{4}$$

**Proposed Temporal Metric Learning:** For location, the visual features change significantly due to differences in landmarks, resulting in the visual embeddings not maintaining a smooth transition with the distance in location embeddings. For instance, two neighborhoods in the same city might appear markedly different. Therefore, formulating standard contrastive objectives with positives and negatives is appropriate for location-image alignment (Eq 2). However, visual cues vary more gradually over time. Consequently, treating two samples that are temporally close as negatives can significantly hinder learning. This makes defining clear positives and negatives, in regular contrastive loss, challenging for time-image alignment. Motivated by these considerations, we propose a metric-learning based objective called *Temporal Metric Learning*, which accounts for the smooth and cyclic nature of time by matching the similarity of instances according to their temporal distance.

Considering a mini batch with a set of image embeddings $\{V_i \mid i \in \{1, 2, \ldots, B\}\}$ and a set of time embeddings $\{T_i \mid i \in \{1, 2, \ldots, B\}\}$. Instead of defining a set of positive and negative pairs for each embedding, we assign soft targets defined by a distance function between the time associated to the image and to the time embeddings. Since months and hours are cyclical, using the regular $\ell_2$ distance between two normalized month-hour pairs $(\theta_i, \phi_i)$ and $(\theta_j, \phi_j)$ in an Euclidean space can result in overestimated distance values as shown in Figure. 2(b). Instead, we map the points into a toroidal manifold, now the new distance $\delta_{i,j}$ becomes:

$$\delta_{i,j} = \sqrt{\min(1 - |\theta_i - \theta_j|, |\theta_i - \theta_j|)^2 + \min(1 - |\phi_i - \phi_j|, |\phi_i - \phi_j|)^2}. \tag{5}$$

Then, for each anchor image embedding $V_i$, we compute a vector $\boldsymbol{p}_i$ with the normalized cosine similarity scores with respect to all the time embeddings $T_j$ in the batch. Similarly, we also compute another vector $\boldsymbol{q}_i$ with the normalized time distances between the anchor time and the other times in the batch.

$$\begin{aligned}
\boldsymbol{p}_i &= \left\{ p_j \mid p_j = \frac{\exp\left(V_i \cdot T_j / \tau\right)}{\sum_{k=1}^{B} \exp\left(V_i \cdot T_k / \tau\right)}, \quad j \in \{1, \ldots, B\} \right\}, \\
\boldsymbol{q}_i &= \left\{ q_j \mid q_j = 1 - \frac{\exp\left(\delta_{i,j}\right)}{\sum_{k=1}^{B} \exp\left(\delta_{i,k}\right)}, \quad j \in \{1, \ldots, B\} \right\}.
\end{aligned} \tag{6}$$

Since all the elements of $\boldsymbol{p}_i$ and $\boldsymbol{q}_i$ add up to one, they essentially represent probability mass functions. Thus, we can define the image-time contrastive loss by using the mean KL-divergence for all samples in the batch $\mathcal{L}_{time} = \frac{1}{B} \sum_{i=1}^{B} \text{KL}(\boldsymbol{p}_i || \boldsymbol{q}_i)$. Our overall training objective is adding both alignment objectives $\mathcal{L}_{time}$ and $\mathcal{L}_{loc}$.

**Inference:** After training the model, the image, location and time modalities are aligned into the same feature space. Thus, in order to predict the location and time-of-capture of a query image $I^Q$, we need to compute the cosine similarity between its embedding with the embeddings from the location and time galleries. Then, the predicted location and time are the samples of the galleries with the highest cosine similarity. Please see Supplementary Section B.

## 4 EXPERIMENTS

**Datasets and evaluation details:** For training, we use two existing datasets: MediaEval Placing Tasks 2016 (MP-16) (Larson et al., 2017) and Cross-View Time (CVT) (Salem et al., 2020). MP-16 consists of 4.72 Million images from Flickr annotated only with GPS coordinates. CVT originally consists of 206k geo-tagged smartphone pictures from the Yahoo Flickr Creative Commons 100 Million Dataset (Thomee et al., 2016) and 98k images from static outdoor webcams of the SkyFinder Dataset (Mihail et al., 2016).

Geo-localization performance is evaluated by measuring the geodesic distance between the real and predicted GPS coordinates, and then computing the ratio of images that are correctly predicted given five different distance thresholds (1 km, 25 km, 200 km, 750 km, and 2500 km). Time prediction performance is evaluated by measuring the mean absolute month ($E_m$) and hour ($E_H$) errors between the real and predicted times. We also report an overall **Time Prediction Score** ($TPS$) to select a single best time model. We compute the $TPS$ based on our proposed cyclical time distance using the following equation:

$$TPS = 1 - \sqrt{\frac{\widetilde{E}_m^2 + \widetilde{E}_H^2}{2}}, \tag{7}$$

where $\widetilde{E}_m$ and $\widetilde{E}_H$ are the normalized time errors. Both have a range between $[0, 1]$; 0 represents a perfect prediction and 1 indicates a maximum error of half the period of the respective time scale. Please refer to Supplementary Sections C to E for more details about the dataset, architecture and training protocol.

### 4.1 COMPARISON WITH STATE-OF-THE-ART METHODS IN TIME PREDICTION

We present our time prediction results in Table 1. Given the significant reproducibility challenges in previous time prediction works (see Supplementary Section F), we selected two previous methods as baselines that provide a fair comparison with GeoTimeCLIP. The first baseline is the triple encoder architecture by Zhai et al. (2019), which is the most similar to our approach. Although the authors do not provide the source code, model weights, or exact dataset splits necessary for replication, they offer a comprehensive description of their protocols, which we strive to follow closely. The second baseline is Padilha et al. (2022), chosen due to its more recent publication, the availability of source code, and its use of the CVT dataset for training, similar to our work. The main challenge, however, is that they do not provide the cross-camera split of CVT, which is the main focus of our evaluation. Additionally, while they only present qualitative results for time prediction, we extend this to quantitative evaluation to facilitate comparison. Both Zhai et al. (2019) and Padilha et al. (2022) use images and geo-locations as inputs, whereas our method relies solely on images. Furthermore, we introduce a third, more robust baseline, TimeCLIP, which uses only the image and time encoders from GeoTimeCLIP. This model is akin to GeoCLIP but focuses exclusively on time prediction. Our experiments demonstrate that GeoTimeCLIP achieves lower errors for month and hour predictions compared to all baselines, without the need for additional metadata. Moreover, training a model for both time prediction and geo-localization simultaneously results in richer time representations. In Figure 3, we provide two sample predictions from our model on images from the SkyFinder dataset, showcasing its capabilities in both time prediction and geo-localization.

### 4.2 ABLATION STUDIES

In the following section we conduct further experiments to show the effect of our proposed Temporal Metric Loss, as well as the robustness of GeoTimeCLIP against limited training data and noisy time annotations. We refer the reader to Supplementary Section G.1 for additional experiments using different time encoders and ToY scales.

**Effect of the loss function for time prediction:** We also experiment with different loss functions for time. The first approach uses a simple CLIP-based loss (Radford et al., 2021). Second, we tried the same loss used in the geo-localization contrastive loss with a dynamic queue and a false negative mask to remove the samples close to the anchors, given a specified threshold. Third, we use the Rank-N-Contrast loss proposed by Zha et al. (2024), which is specifically designed to tackle regression problems by contrasting samples against each other by ranking them according to a

Table 1: **Zero-shot time prediction** on the *unseen* cameras of the SkyFinder dataset. The * indicates methods that we trained ourselves, closely adhering to the protocols outlined by the original authors.

| Method | Month Error | Hour Error | TPS |
|---|---|---|---|
| Random guess | 2.97 | 3.96 | 57.93 |
| Zhai et al. (2019)* | 2.46 | 3.18 | 65.48 |
| Salem et al. (2022)* | 1.62 | 3.61 | 71.42 |
| TimeCLIP | 1.52 | 2.84 | 75.49 |
| **GeoTimeCLIP** | **1.40** | **2.72** | **77.00** |

Table 2: **Ablations** for time prediction performance with different loss functions. L2 refers to the euclidean distance, while cyclic refers to the the distance over the toroidal manifold.

| Loss Function | Month Error | Hour Error | TPS |
|---|---|---|---|
| CLIP | 1.71 | 3.51 | 71.12 |
| RnC | 1.87 | 2.96 | 71.89 |
| SimCLR | 1.50 | 3.4 | 73.28 |
| TML (L2) | 1.56 | **2.69** | 75.73 |
| **TML (Cyclic)** | **1.40** | 2.72 | **77.00** |

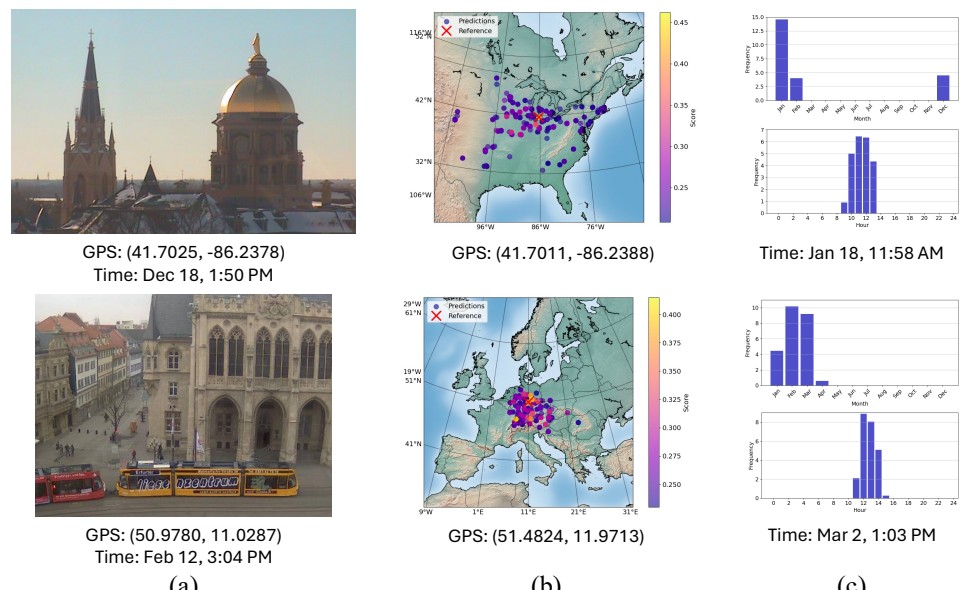

GPS: (41.7025, -86.2378)
Time: Dec 18, 1:50 PM

GPS: (41.7011, -86.2388)

Time: Jan 18, 11:58 AM

GPS: (50.9780, 11.0287)
Time: Feb 12, 3:04 PM

GPS: (51.4824, 11.9713)

Time: Mar 2, 1:03 PM

(a)          (b)          (c)

Figure 3: (a) Sample images for two cameras of the SkyFinder test set with the ground truth location and capture time. (b) Spatial distribution of the predicted GPS coordinates colored by the cosine similarity between the location and image embeddings. (c) Temporal distribution of the predicted month and hour, weighted by the cosine similarity between the time and image embeddings.

distance metric. Lastly, we present results using our novel loss function replacing the cyclic time distance with the regular $\ell_2$ distance. The results, show in Table 2, show that our proposed loss significantly outperforms all other losses in month prediction, and also outperforms all but the variant without cyclic distance on hour prediction. This small difference might me attributed to the fact that the dataset is biased towards daytime. Thus, hours don't see a major benefit from using a cyclic distance.

Table 3: **Ablations** for robustness to limited data for time-prediction.

| Data Availability | Month Error | Hour Error | TPS |
|---|---|---|---|
| 100% | 1.40 | 2.72 | 77.00 |
| 50% | 1.69 | 2.83 | 74.02 |
| 10% | 1.70 | 2.94 | 73.51 |
| 5% | 1.89 | 2.86 | 72.07 |

Table 4: **Ablations** for robustness to label noise for time-prediction.

| Label Noise ($\sigma$) | Month Error | Hour Error | TPS |
|---|---|---|---|
| 0 | 1.40 | 2.72 | 77.00 |
| 1 | 1.52 | 2.71 | 76.00 |
| 2 | 1.75 | 2.74 | 73.81 |
| 3 | 2.16 | 2.72 | 69.92 |

**Experiments on the robustness of GeoTimeCLIP:** Existing publicly available datasets with time-of-capture metadata either have a small number of samples or noisy timestamps. Furthermore, even fewer datasets have fully open sources licenses, restricting their use to research purposes. Taking into account these issues, we explore the robustness of our model on settings with limited data availability as well as injected timestamp noise. For data availability, we progressively remove training samples from the geo-localization and time prediction datasets going from 100% to 5% in four steps. Similarly, for the noisy setting, we inject Gaussian noise into the time labels with increasing standard deviations, going from 0 to 3 months and hours. In Tables 3 and 4, we show that our model is surprisingly robust against both types of perturbations. Even with only 10% of the data (12.5k samples), the model is still able to learn meaningful time representations, with increased time errors of 0.3 months 0.22 hours. In the noisy labels setting, GeoTimeCLIP is able to handle noise levels up until $\sigma = 2$ hours and months without a major drop in performance. For setting with $\sigma = 3$, the error of the month prediction task increases significantly, but the time error stays constant.

## 4.3 COMPOSITIONAL IMAGE RETRIEVAL

We also explore the capability of our model for image retrieval: given a pair of query GPS coordinates $G^Q$ and timestamps $U^Q$ from the SkyFinder test set, we want to retrieve an image that corresponds to the specific location and time. We explore a simple method by averaging the query location $L^Q$ and time $T^Q$ embeddings to generate a multimodal spatio-temporal embedding, inspired by similar approaches by Shvetsova et al. (2022a) and Swetha et al. (2023) between video, audio and text modalities. The spatio-temporal embedding is then compared against a gallery of image embeddings and we select the one with the highest cosine similarity. We show qualitative results in Figure 4.

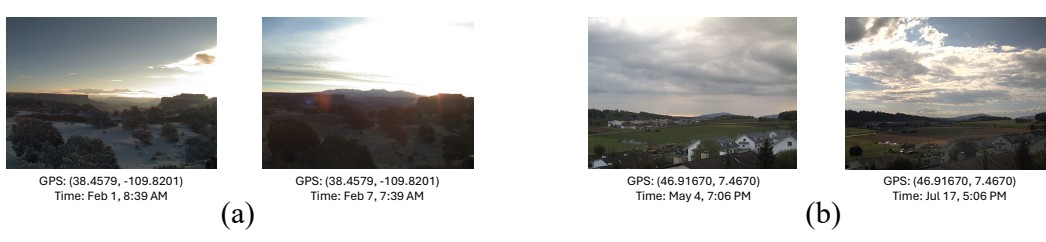

Figure 4: Illustrating *compositional $L + T \rightarrow I$* retrieval with GeoTimeCLIP. Each example Left: query location and query time, showing actual image. Right: retrieved image for given query location and time.

## 4.4 QUALITATIVE RESULTS USING TEXT QUERIES

We also investigate the ability of GeoTimeCLIP of using the pretrained CLIP text encoder to retrieve times and locations mentioned in the text. For this task, we follow GeoCLIP's approach and replace the image backbone by a text backbone, keeping the trained MLP, location encoder and time encoder. For each text, we create a text embedding, pass it through the MLP and compare it agains the location and time galleries. We then create spatial and temporal distributions of the top retrieved samples for each modality, as shown Figure 5, where we see that not only is our model able to pinpoint the location with high accuracy, but also create meaningful time distributions to words such as "summer" and "evening" that do not explicitly mentioned the time.

## 4.5 COMPARISON WITH STATE-OF-THE-ART METHODS IN GEO-LOCALIZATION

Table 5 presents the geo-localization performance of our model compared to state-of-the-art methods. Overall, GeoTimeCLIP delivers competitive results against state-of-the-art models like GeoCLIP and PIGEOTTO on the Im2GPS3k dataset. While Img2Loc achieves the lowest errors on Im2GPS3k, its performance is highly dependent on the choice of MLLM, as demonstrated by the significant difference between LLaVA and GPT-4V. Notably, GeoTimeCLIP outperforms the LLaVA-based Img2Loc variant, despite using at least 15 times fewer parameters.

On the more challenging GWS15k dataset, GeoTimeCLIP achieves state-of-the-art performance at the 1 km scale and ranks second across all other scales. Unlike Im2GPS3k, which consists

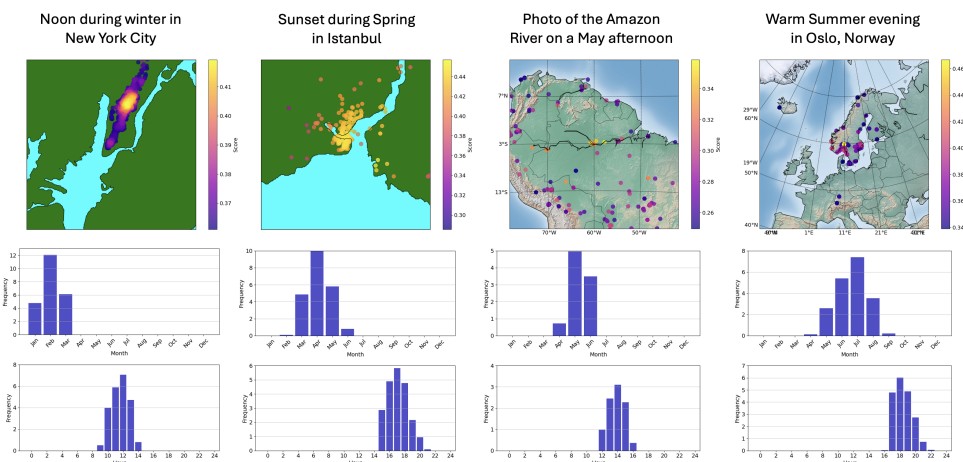

Figure 5: Qualitative examples of geo-localization and time-of-capture prediction using text queries. GeoTimeCLIP is capable of providing good estimates of the time and GPS coordinates to each of the text queries, even when time is not explicitly specified.

Table 5: **Geo-localization results** on Im2GPS3k & GWS15k datasets, reported on the ratio of samples that are correctly predicted under different distance thresholds. GeoTimeCLIP achieves comparable performance to other state-of-the-art methods that focus exclusively on geo-localization for all scales.

| Method | Im2GPS3k | | | | | GWS15k | | | | |
| --- | --- | --- | --- | --- | --- | --- | --- | --- | --- | --- |
| | Street 1 km | City 25 km | Region 200 km | Country 750 km | Continent 2500 km | Street 1 km | City 25 km | Region 200 km | Country 750 km | Continent 2500 km |
| [L] kNN, sigma=4 (Vo et al., 2017) | 7.2 | 19.4 | 26.9 | 38.9 | 55.9 | - | - | - | - | - |
| PlaNet (Weyand et al., 2016) | 8.5 | 24.8 | 34.3 | 48.4 | 64.6 | - | - | - | - | - |
| CPlaNet (Seo et al., 2018) | 10.2 | 26.5 | 34.6 | 48.6 | 64.6 | - | - | - | - | - |
| ISNs (Muller-Budack et al., 2018) | 10.5 | 28 | 36.6 | 49.7 | 66.0 | 0.1 | 0.6 | 4.2 | 15.5 | 38.5 |
| Translocator (Pramanick et al., 2022) | 11.8 | 31.1 | 46.7 | 58.9 | 80.1 | 0.5 | 1.1 | 8.0 | 25.5 | 48.3 |
| GeoDecoder (Clark et al., 2023) | 12.8 | 33.5 | 45.9 | 61 | 76.1 | 0.7 | 1.5 | 8.7 | 26.9 | 50.5 |
| GeoCLIP (Vivanco Cepeda et al., 2024) | 14.1 | 34.5 | 50.7 | 69.7 | 83.8 | 0.6 | 3.1 | 16.9 | 45.7 | 74.1 |
| PIGEOTTO (Haas et al., 2024) | 11.3 | **36.7** | **53.8** | **72.4** | **85.3** | 0.7 | **9.2** | **31.2** | **65.7** | **85.1** |
| Img2Loc (LLaVA) (Zhou et al., 2024) | 8.0 | 23.4 | 29.9 | 40.1 | 51.1 | - | - | - | - | - |
| Img2Loc (GPT-4V) (Zhou et al., 2024) | 17.1 | 45.1 | 57.9 | 72.9 | 84.9 | - | - | - | - | - |
| GeoTimeCLIP | **14.4** | 33.1 | 49.4 | 68.4 | 83.5 | **0.9** | 3.48 | 17.1 | 47.2 | 74.6 |

mainly of images of famous landmarks, requiring models to memorize specific locations, GWS15k includes images randomly distributed across the globe. The majority of these images depict generic locations, offering minimal location-specific information beyond broader contextual cues, such as architectural styles and vegetation. This suggests that GeoTimeCLIP may be learning more generalizable location features compared to GeoCLIP, potentially at the expense of its ability to memorize the visual characteristics of specific landmarks. On this dataset, PIGEOTTO obtains the best overall performance, likely benefiting from the extensive use of additional metadata used during training.

## 5 CONCLUSION

We introduce GeoTimeCLIP, a novel framework for jointly predicting the time and location of an image using a retrieval approach. GeoTimeCLIP not only shows competitive performance compared to state-of-the-art geo-localization models but also introduces the capability of precise time-of-capture predictions. A key innovation of our approach is the novel temporal metric loss, which significantly outperforms traditional contrastive losses in time prediction tasks. Furthermore, our results demonstrate that GeoTimeCLIP extends beyond standard prediction and geo-localization tasks. It supports additional functionalities like compositional image retrieval, as well as text-to-location and text-to-image retrieval, indicating a profound understanding of the interplay between images, locations, and time.

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

## A    APPENDIX

We organize the supplementary as follows. In Section B we present the inference pipeline of our framework, followed by training and implementation details in Section C and dataset details in Section E.

## B    MODEL INFERENCE

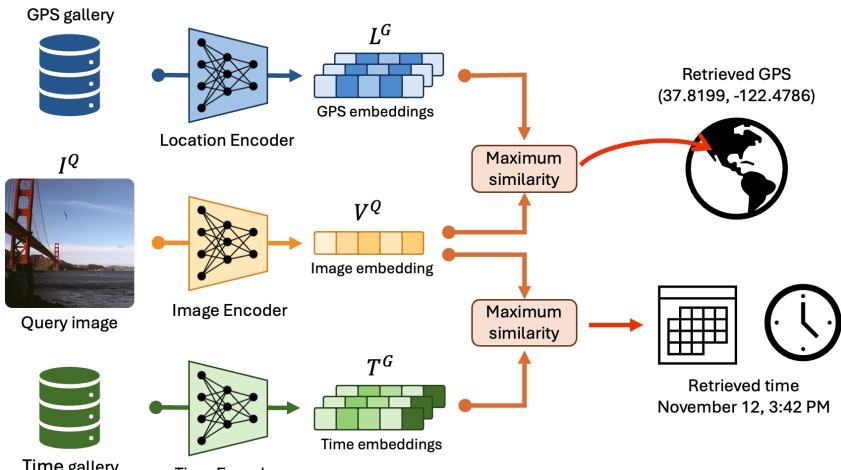

Figure 6: Our framework consists of a model that can predict both the location and capture-time of an image at the same time using a retrieval approach. Given a query image $I^Q$, a gallery of GPS coordinates and a gallery of timestamps, GeoTimeCLIP maps the three modalities into a shared feature space using an image, location and time encoder. The query image embedding $V^Q$ is compared against a set of location embeddings $L^G$ and time embeddings $T^G$. The GPS and timestamp with the highest cosine similarity are selected as the predictions of our model.

Figure 6 represents the overview of our approach.

## C    IMPLEMENTATION DETAILS

Following GeoCLIP, the backbone of the image encoder is a pretrained ViT-L/14 from CLIP and the MLP consists of two fully connected layers with ReLU activation function and dimensions 768 and 512 respectively. We use the same architecture for the time and location encoders as GeoCLIP. Both employ three RFF positional encoding layers, mapping the 2-dimensional GPS to a vector with 512 dimensions. The standard deviation values used to sample the RFF are $\sigma_i \in \{2^0, 2^4, 2^8\}$. The MLPs from the time and location encoder have three hidden layers with 1024 dimensions and a projection layer to map the final embeddings into a feature space of 512 dimensions. In the location encoder, we use a dynamic queue that stores the last 4096 seen locations, but we don't use it for time. The GPS coordinates and times are augmented by adding Gaussian noise with standard deviation of 150 meters for the in-batch GPS, 1500 meters for the GPS queue, 0.15 months and 0.15 hours for time. We perform two augmentations for each image in the training set using random resized crops of size 224, random horizontal flipping and image normalization.

## D    TRAINING PROTOCOL

GeoTimeCLIP is trained for 20 epochs using a cosine decay scheduler, with learning rate values ranging from $\alpha_{max} = 3 \times 10^{-5}$ to $\alpha_{min} = 3 \times 10^{-7}$. We use Adam optimizer with coefficients $\beta_1 = 0.9$, $\beta_2 = 0.999$ and $\ell_2$ penalty of $1 \times 10^{-6}$. For the contrastive losses, we use two learnable temperature parameters that are optimized during training. The batch size $B$ is set to 512 for all experiments, and the models are trained on a machine with 12 CPU cores and a NVIDIA RTX A6000 GPU.

# E    DATASET DETAILS

We apply two filters to remove samples from the CVT that don't provide meaningful temporal information. In particular, we remove all night-time and indoor images, since they often have inconsistent temporal cues. To remove night images, we estimate the sunrise and sunset times from the date, latitude and longitude using the General Solar Position algorithm, and remove all samples before sunrise or after sunset. Then, for indoor images, we leverage a CNN model pretrained on the Places365 Dataset Zhou et al. (2017). In general, night images often have inconsistent artificial lighting, more noise or specialized cameras such as night vision. Indoor images also have artificial lighting and controlled temperature, making it difficult to estimate the time or date.

Regarding the levels of noise in the dataset, the SkyFinder subset consists of images with accurate time estimates, since they were collected from calibrated outdoor webcams. However, we observed that CVT has a moderate amount of noisy labels. Thus in order to train a model that can accurately predict the time, we need both datasets in the training set.

For evaluating the models, we employ a subset of unseen SkyFinder cameras, as well as two geo-localization datasets used by other state-of-the-art methods for evaluation: Im2GPS3k and GWS15k. Similar to GeoCLIP, we create a 100k GPS gallery to evaluate the model on Im2GPS3k, a 500k GPS gallery for GWS15k, and a 100k time gallery for the SkyFinder test set. The GPS galleries are created by sampling GPS coordinates from the MP-16 dataset, while the time gallery is created by sampling times from the combined CVT and SkyFinder training sets.

# F    LIMITATIONS AND REPRODUCIBILITY CHALLENGES OF EXISTING TIME PREDICTION METHODS

Most previous time prediction methods suffer from a lack of standardization in their training and evaluation protocols. For instance, Zhai et al. (2019) use subsets of the AMOS (Jacobs et al., 2007; 2009) and YFCC100M (Thomee et al., 2016) datasets, without providing the source code or exact dataset splits necessary to replicate their experiments. Additionally, their time prediction evaluation relies on cumulative error plots, but they do not provide a single numerical value summarizing the performance of their model. Similarly, while Salem et al. (2020) and Padilha et al. (2022) offer datasets and code, they do not include the cross-camera split for zero-shot time prediction—a more challenging and informative evaluation protocol that we adopt. Moreover, their results are presented only qualitatively, though they can be adapted for obtaining quantitative results. Salem et al. (2022) also omit critical details such as dataset splits for the SkyFinder dataset, do not clarify whether their results corresponds to same- or cross-camera evaluation, and fail to provide the source code for replication. Their use of top-$k$ accuracy as an evaluation metric further complicates direct comparisons. Notably, none of these methods, with the exception of Salem et al. (2020) and Padilha et al. (2022), compare their time prediction performances against each other, and even these comparisons are only qualitative. Other time prediction approaches, including those by Tsai et al. (2016), Li et al. (2017), Lalonde et al. (2012), and Hold-Geoffroy et al. (2017), also face similar challenges, such as a lack of available source code, missing datasets, or datasets that are no longer hosted online. This lack of standardization prevents consistent benchmarking across different methods.

# G    ADDITIONAL ABLATIONS

## G.1    TIME-OF-YEAR SCALE

Time-of-year (ToY) can be represented at either a monthly or daily scale. In practice, the choice of time scale should not significantly affect the results, as the value is normalized before being passed to the time encoder, $\mathcal{T}(\cdot)$. However, two approaches are available. The first approach converts the integer month $m_i$ and day $d_i$ into a real-valued month, normalized over a 12-month period, as shown in Equation 3. The second approach represents ToY as the number of days elapsed since the start of the year, normalized over 365 days (assuming no leap years in our dataset). This representation is defined as:

$$\theta_i = \frac{1}{365}\left(d_i - 1 + \sum_{k=1}^{i} \mathcal{D}(m_k - 1)\right),$$

Table 6: **Ablations** for time prediction performance using monthly and daily scales.

| ToY Scale | Month Error | Hour Error | TPS |
|---|---|---|---|
| daily | 1.53 | 2.71 | 75.91 |
| monthly | 1.40 | 2.72 | 77.00 |

Table 7: **Ablations** for time prediction performance using different time encoders.

| Time encoder | Month Error | Hour Error | TPS |
|---|---|---|---|
| Circular decomp. | 1.59 | 2.86 | 74.80 |
| Time2Vec | 1.56 | 2.62 | 75.99 |
| RFF | 1.40 | 2.72 | 77.00 |

where $\mathcal{D}(m_k)$ is the number of days in month $m_k$, and it is assumed that $\mathcal{D}(0) = 0$. In Table 6, we empirically show that using a monthly scale for ToY representation results in slightly better performance. However, we attribute this improvement to statistical noise rather than the time representation method itself.

### G.2 TIME REPRESENTATION

Motivated by the cyclical nature of time, Mac Aodha et al. 2019 used **circular decomposition** to wrap the temporal input to their geographical prior encoder, resulting in similar embeddings for dates that are close to the start and end of the year, such as December $31^{st}$ and January $1^{st}$. To achieve this, for each dimension $l$ of the temporal input $x$, they perform the mapping $[\sin(\pi x^l), \cos(\pi x^l)]$, resulting in two numbers for each dimension.

**Time2Vec** (Kazemi et al., 2019) is a method for encoding time that captures both periodic and non-periodic patterns. It transforms scalar time values into a vector of size $k+1$. The first element models linear, non-periodic trends, while the remaining elements are defined by a periodic activation function (e.g., sine), capturing repeating temporal behaviors like daily or weekly cycles. The representation is defined as:

$$t2v(\tau)[i] = \begin{cases} \omega_i \tau + \phi_i & \text{if } i = 0, \\ F(\omega_i \tau + \phi_i) & \text{if } 1 \leq i \leq k, \end{cases}$$

where $F$ is a periodic activation function, typically $\sin$, and $\omega_i$ and $\phi_i$ are learnable parameters representing the frequency and phase shift, respectively.

## H ADDITIONAL QUALITATIVE RESULTS

We show additional qualitative results of our method in figures 7 and 8. We include a failure case, on the last row of figure 8, where the time error is high because of the presence of fog in the image.

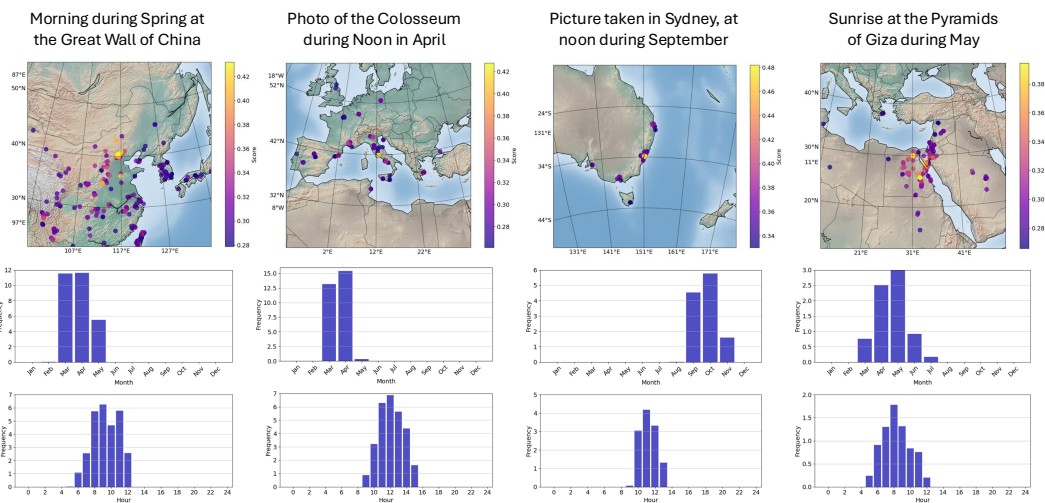

Figure 7: Additional qualitative examples of geo-localization and time-of-capture prediction using text queries.

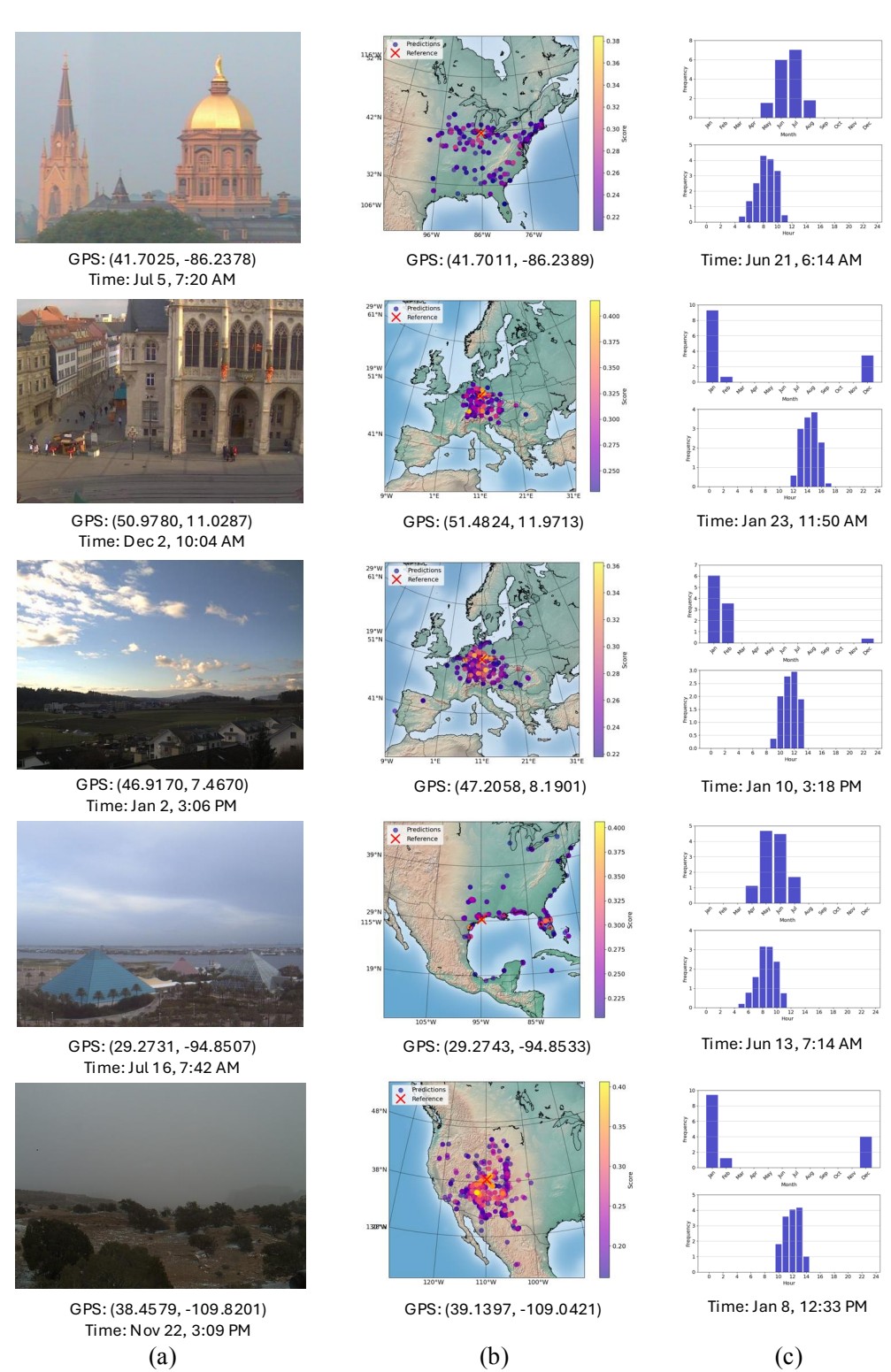

Figure 8: (a) Additional sample predictions for three cameras of the SkyFinder test set with the ground truth location and capture time. (b) Spatial distribution of the predicted GPS coordinates colored by the cosine similarity between the location and image embeddings. (c) Temporal distribution of the predicted month and hour, weighted by the cosine similarity between the time and image embeddings.

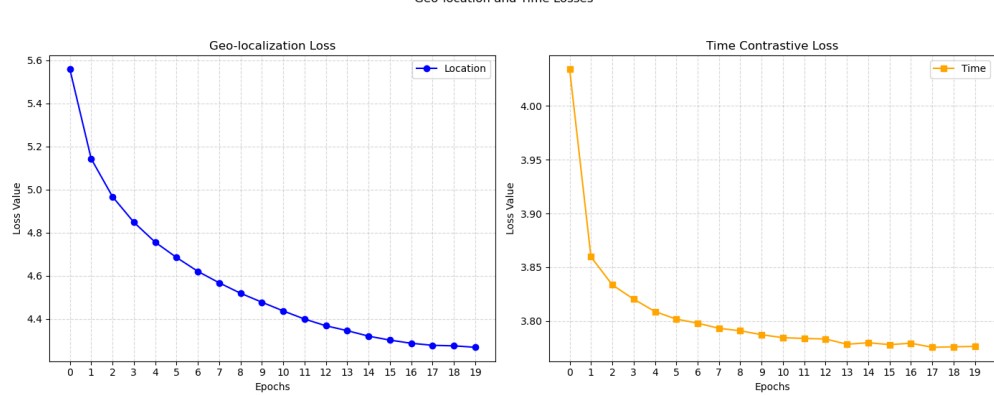

Figure 9: Convergence of the geo-localization loss ($\mathcal{L}_{loc}$) and the Time Metric Loss ($\mathcal{L}_{time}$) during training.

## I  TRAINING STABILITY

Maintaining training stability requires batches with consistent temporal and geographical distributions, which can be challenging with smaller batch sizes. To address this, we use batches of 512 randomly sampled images that span diverse locations, hours, and months. As shown in Figure 9, our batching strategy ensures smooth convergence of both the geo-localization loss ($\mathcal{L}_{loc}$) and the time metric loss ($\mathcal{L}_{time}$).

## J  GEOTIMECLIP PREDICTIONS ACROSS DIFFERENT LATITUDES

Figure 10 compares time prediction examples from GeoTimeCLIP and TimeCLIP, a baseline model trained solely with the visual ($\mathcal{V}$) and temporal ($\mathcal{T}$) encoders. The results suggest that TimeCLIP struggles more with hour predictions at higher latitudes (40° to 70°) compared to GeoTimeCLIP. In contrast, at moderate latitudes (-40° to 40°), both models exhibit more consistent hour prediction errors, though GeoTimeCLIP demonstrates superior performance in month prediction.

## K  SIMILARITY MAPS

To gain insights into the features that GeoTimeCLIP leverages for its geo-temporal predictions, we create attention maps by computing the cosine similarity between the CLIP class embedding and patch embeddings after passing through the image encoder's MLP. These attention maps allow us to identify which regions of an image are most relevant to the model's predictions.

Figure 11 showcases attention maps for three example images. The highlighted regions correspond to the features that GeoTimeCLIP considers most significant when predicting time and location:

- Sky and Time of Day (Figure 11a): In this example, the attention is primarily focused on the sky. The color and lighting conditions in the sky serve as strong indicators of the time of day, providing critical cues for time-of-day predictions.

- Foliage and Seasonal Changes (Figure 11b): In the second example, the model shifts its focus to the leaves and branches of trees. Seasonal variations in foliage may play a significant role in estimating the time-of-year.

- Architectural Features and Geographical Context (Figure 11c): In urban scenes, the model's attention centers on buildings. Architectural styles can provide clues about the geographical location of the image.

**Latitude range:**                                                    **40° to 70°**

|  | | | |
|---|---|---|---|
| *Ground truth* | Nov 16, 13:48:02 | Oct 19, 16:56:37 | Apr 24, 16:58:49 |
| *TimeCLIP* | May 08, 15:07:28 **(TPS=32.07%)** | Apr 15, 13:13:52 **(TPS=27.26%)** | Oct 15, 12:55:12 **(TPS=28.86%)** |
| *GeoTimeCLIP* | Nov 26, 14:15:44 **(TPS=95.22%)** | Aug 26, 15:02:56 **(TPS=76.30%)** | Mar 07, 14:45:20 **(TPS=77.29%)** |

**Latitude range:**                                                    **10° to 40°**

|  | | | |
|---|---|---|---|
| *Ground truth* | Jun 20, 18:02:41 | Jun 14, 15:44:47 | May 26, 13:51:08 |
| *TimeCLIP* | Dec 24, 13:37:52 **(TPS=25.86%)** | Dec 08, 12:20:32 **(TPS=28.85%)** | Nov 27, 13:26:24 **(TPS=29.96%)** |
| *GeoTimeCLIP* | May 23, 15:59:12 **(TPS=83.70%)** | Jun 10, 13:41:20 **(TPS=87.78%)** | Apr 26, 13:46:40 **(TPS=88.52%)** |

**Latitude range:**                                                    **-10° to 10°**

|  | | | |
|---|---|---|---|
| *Ground truth* | May 18, 12:07:50 | May 25, 08:19:05 | Oct 31, 15:42:46 |
| *TimeCLIP* | Nov 20, 12:44:00 **(TPS=30.20%)** | Dec 22, 11:10:40 **(TPS=37.61%)** | May 17, 13:54:40 **(TPS=34.88%)** |
| *GeoTimeCLIP* | Mar 19, 13:37:52 **(TPS=75.18%)** | Apr 07, 08:24:00 **(TPS=81.44%)** | Aug 21, 13:52:48 **(TPS=70.57%)** |

**Latitude range:**                                                    **-40° to -10°**

|  | | | |
|---|---|---|---|
| *Ground truth* | May 05, 12:14:06 | Feb 10, 13:54:42 | Apr 14, 14:18:05 |
| *TimeCLIP* | Nov 21, 13:38:24 **(TPS=35.10%)** | Aug 23, 14:02:56 **(TPS=33.99%)** | Aug 18, 14:26:56 **(TPS=30.65%)** |
| *GeoTimeCLIP* | Apr 30, 12:53:36 **(TPS=95.67%)** | Mar 23, 13:17:52 **(TPS=83.12%)** | Jun 20, 14:11:28 **(TPS=74.02%)** |

Figure 10: Sample predictions where GeoTimeCLIP outperforms the TimeCLIP baseline across different latitudes.

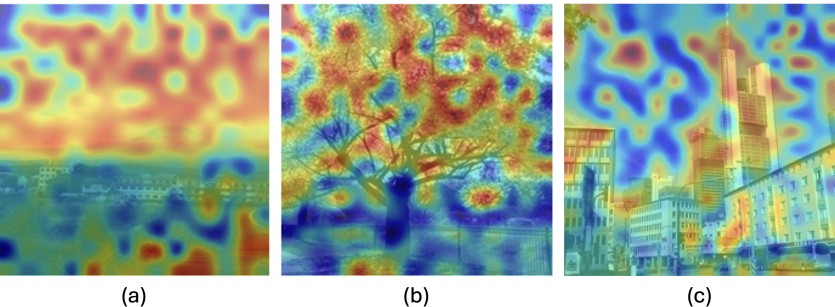

(a)  (b)  (c)

Figure 11: Attention maps computed from GeoTimeCLIP's image embeddings, highlighting geo-temporal features like the sky, foliage, and buildings.

## L  ANALYSIS OF THE LEARNED EMBEDDING SPACE

One of the key motivations for GeoTimeCLIP is to align images, time, and location in a shared multi-modal embedding space. This approach is inspired by prior works like GeoCLIP (Vivanco Cepeda et al., 2024), SatCLIP (Klemmer et al., 2023), and CSP (Mai et al., 2023), which embed images and GPS coordinates in shared spaces, as well as methods like ImageBind Girdhar et al. (2023), LanguageBind (Zhu et al., 2023), and Everything At Once (Shvetsova et al., 2022b), which align multiple modalities such as images, text, videos, and audio. Our work extends this idea to include temporal information, showing in Table 1 that aligning these three modalities leads to improved time prediction performance compared to using only images and time.

To explore the relationships between these modalities in the learned embedding space, we performed Principal Component Analysis (PCA) on the embeddings. While PCA has limitations in fully capturing the underlying structure of high-dimensional spaces, the results provide interesting qualitative insights. Figure 12(a) presents the distributions of image, time, and location embeddings, appearing in different subspaces. Figures 12(b) and 12(c) show more details about the relationship between image and time embeddings. The image embeddings are clustered in the center, surrounded by time embeddings. Notably, the directions of hours and months are well-defined: months are radially distributed, while hours are linearly distributed in a perpendicular direction. For location embeddings (Figures 12(d-e)), even though the patterns are less pronounced, the embeddings at different latitudes and longitudes still form distinct clusters in the feature space.

## M  ABLATIONS WITH DIFFERENT IMAGE BACKBONES

To evaluate the impact of different image embeddings on time prediction performance, we conducted ablation studies using three backbones: DINOv2-L (Oquab et al., 2023), OpenCLIP ViT-G (Ilharco et al., 2021), and OpenAI's original CLIP ViT-L (Radford et al., 2021). For these experiments, we used the TimeCLIP model, which incorporates only the image and time encoders. The results, summarized in Table 8, indicate that OpenAI's CLIP ViT-L achieves the lowest errors for both hours and months, as well as the highest Time Prediction Score (TPS).

| Backbone | Parameters | Month Error | Hour Error | TPS |
|---|---|---|---|---|
| DINOv2 ViT-L/14 | 0.3B | 2.10 | 3.25 | 68.71 |
| OpenCLIP ViT-G/14 | 1.8B | 1.57 | 2.94 | 74.65 |
| **OpenAI CLIP ViT-L/14** | **0.2B** | **1.52** | **2.84** | **77.54** |

Table 8: Comparison of time prediction performance using different backbones.

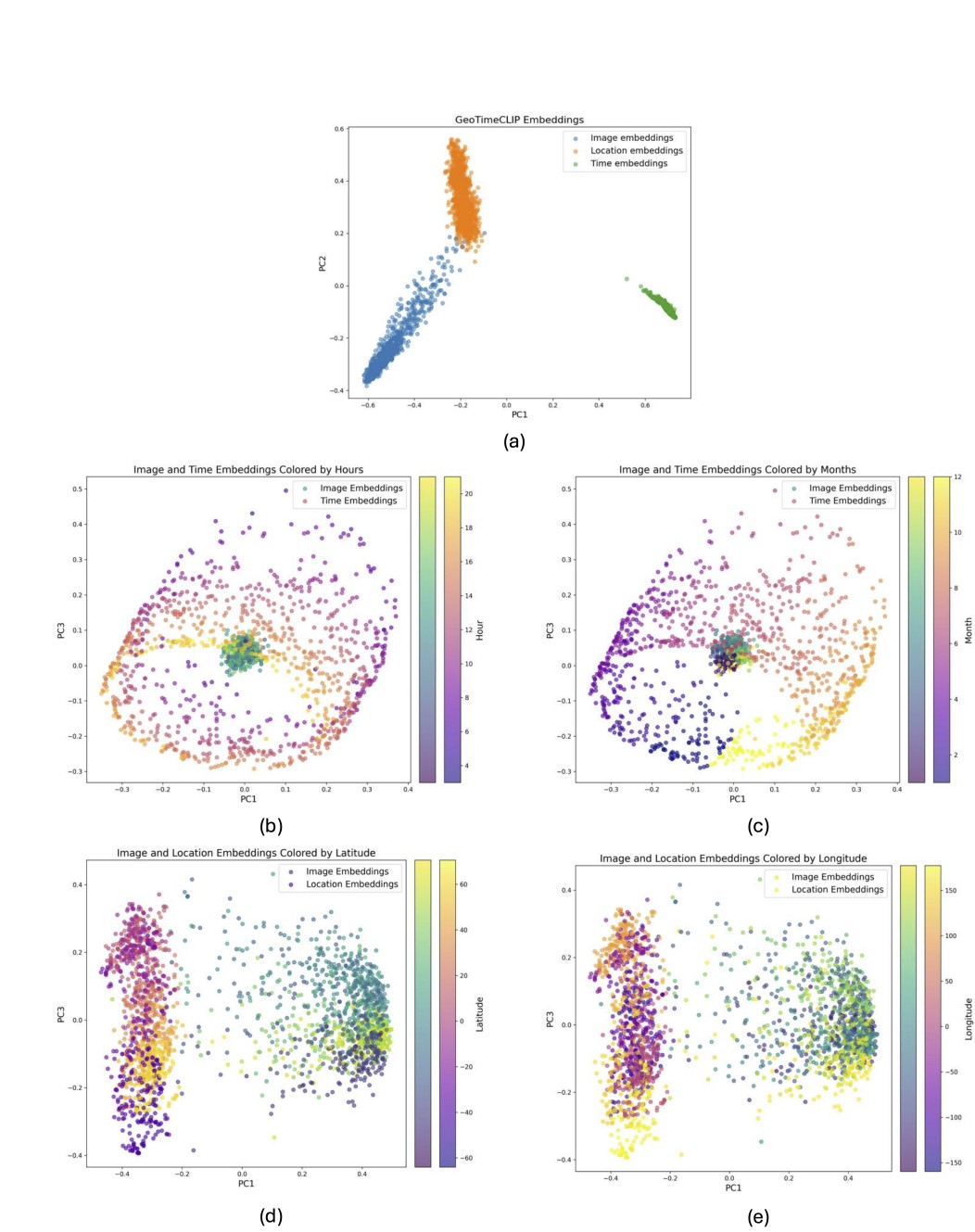

Figure 12: PCA plots of the embedding spaces in GeoTimeCLIP. (a) Distribution of the image, time and location embeddings. (b)-(c) Distribution of the image and time embeddings, colored by the time-of-day and time-of-year respectively. (c)-(d) Distribution of the image and location embeddings, colored by the latitude and longitude respectively.

