# OpenReview forum: "GeoTimeCLIP: Unveiling the When and Where of Images"
_ICLR.cc/2025/Conference — Submitted to ICLR 2025_

### Official Review · Reviewer_sJvB · 2024-11-02

**Soundness:** 4
**Presentation:** 4
**Contribution:** 4
**Rating:** 6
**Confidence:** 5

**Summary:**

This work proposes a unified embedding for location, time, and a ground-level image. This is similar to recent approaches, such as GeoCLIP, but extends the representation to include time. Briefly, a frozen CLIP encoder is used to represent the image features. A lightweight MLP is used to project this into the shared representation space. Two separate lightweight encoders are used to embed location and time. Sensible choices are made for the input positional encodings for both domains. Image-location similarity is optimized using a fairly standard contrastive loss and Image-time similarity is optimized using a combination of a contrastive loss and a distance loss.

**Strengths:**

- The problem is interesting and breaks new ground relative to recent works on static geospatial embeddings.
- The elements of the approach make sense.
- The discussion of related work is solid.
- The presentation is clear.

**Weaknesses:**

The paper is generally well executed, but I do see a few issues:

1) The problem domain is fairly niche. I don't see that as a major issue given the recent interest in image localization, timestamp estimation, and image embeddings in general.

2) It seems odd to put image and time into the same embedding space since they are distinct concepts. It seems that having two distinct embeddings might facilitate additional applications instead of having the two representations intertwined.
(2a) Along this direction, I would have liked to see some exploration of the learned embedding space. Do space and time happen to live in different subspaces? If so, that would point toward the potential value of just making this a hard constraint.

3) The compositional image retrieval experiment doesn't seem to add much value. It seems like it would have been more interesting to look at what the similarity maps of the embeddings look like for arbitrary locations and times. These embeddings could emphasize the power of the embedding, or highlight areas for improvement.

4) I would have liked to see more details about the re-implementation of Zhai 2019 and Salem 2022. For example, are these using the CLIP encoder or the weaker encoders that were used in the original papers? This is also an issue with Table 5, where some of the difference between the methods could be attributed to the difference in the underlying image encoder, not all of the additional aspects which are the claimed contributions of this work. This concern could easily be addressed by experiments across different backbones (perhaps a weaker and a stronger ImageNet pre-trained model). That should be fairly quick to do given the relatively lightweight nature of this approach.

5) Perhaps I missed it, but it's not clear why the particular approach for the Temporal Metric was selected. There seem to be quite a few variants of this, but only L2 is evaluated as a baseline. The results don't seem particularly conclusive in Table 2.

Minor issues:
a) The bibtex entries for at least a couple of the papers need work: Salem 2022 and Zhai 2019 are missing the venue.

**Questions:**

L248: It took a while to find the number of scales that were used here. I think this is defined on L739 as 3, but it's not associated with the $M$. Is this correct?

Eq 1 Why was this approach taken? Why not feed all features into one MLP? Why not use the concatenation of the MLP outputs?

What is the impact of different image encodings?

---

### Official Review · Reviewer_bG3A · 2024-11-03

**Soundness:** 3
**Presentation:** 3
**Contribution:** 3
**Rating:** 6
**Confidence:** 4

**Summary:**

This paper, GEO TIME CLIP: UNVEILING THE WHEN AND WHERE OF IMAGES, proposes a basic retrieval method named GeoTimeCLIP, which can be used to combine time prediction and geopositioning of images. The study highlights the importance of accurately estimating the time and geographic location of image capture, especially in areas such as digital forensics, ecological research, and social media management. While existing methods typically rely on GPS data for time estimation, GeoTimeCLIP uses contrast learning to align images, time, and location by building a shared multimodal feature space.

**Strengths:**

1. The GeoTimeCLIP method proposed in this paper has significant originality in the joint modeling of time prediction and geographical positioning. By representing time as a month-hour pair and considering the periodicity of time, the authors shed new light on the problem of time prediction. In addition, the proposed measurement learning method based on time difference overcomes the limitation of the traditional contrast learning method in time prediction, and presents an innovative improvement to the existing methods.
2. The quality of the paper is reflected in its methodological rigor and experimental comprehensiveness. The author not only proposed a new model framework, but also verified the effectiveness of the method through a new benchmark test. GeoTimeCLIP excelled in the combined prediction of time and location, going beyond the baseline of optimizing time only and competing with expert-level geolocation methods, demonstrating its high quality research results.
3. The structure of the paper is clear and the logic is rigorous. In the introduction part, the background and importance of the research are clearly expounded, and then the design idea and implementation details of the method are introduced in detail. The use of diagrams and formulas effectively assists readers in understanding complex concepts, making the paper as a whole easy to read and understand.

**Weaknesses:**

1. I recommend that the authors consider establishing a public benchmark dataset containing images from a variety of scenarios and conditions in future work. This would help promote research in this field and provide a valuable resource for comparative studies.
2. The study does not address the effects of different latitudes and climatic conditions on time representation. It would be beneficial to consider these factors, as they could significantly impact the model's performance. Exploring this aspect could enhance the applicability of the model across diverse geographic regions.
3. Although the current model shows improved performance, it lacks interpretability in its decision-making process. I recommend introducing interpretability techniques in future studies to help users understand how the model makes predictions about time and place. This would enhance user trust and provide insights into the model's internal workings.

**Questions:**

1.The authors mention that GeoTimeCLIP is able to predict time and place without additional metadata, but in practical applications, how to ensure its accuracy? In particular, is the performance of the model affected by different geographical locations and seasonal changes?
2.What do the authors think are the potential applications of GeoTimeCLIP's findings in future research? Are there plans to apply the method to other fields or to combine it with other techniques for further research?
3.Does the model have additional requirements for the attributes of the image, such as tones, filters, etc.?

---

### Official Review · Reviewer_j8ew · 2024-11-07

**Soundness:** 3
**Presentation:** 3
**Contribution:** 3
**Rating:** 5
**Confidence:** 5

**Summary:**

The paper learns both  image-time embedding alignment as well as  image-geo embedding alignment in a retrieval way.
Days and years are represented using Random Fourier Features to handle cyclic patterns effectively.
Instead of standard contrastive loss, a novel metric learning approach is used.
Shared embedding space facilitates downstream tasks, such as compositional and text-based retrieval.

**Strengths:**

- Predict both time and geo-info jointly.
- The new time representation method for both month and hour is really good: normalize the month and hour in the same range. In this way, time values are converted as a pair of continuous numbers.
- The Time Prediction Score provides a good evaluation taking both hour and month into account.
- The proposed temporal metric learning sounds reasonable.

**Weaknesses:**

I have two concerns about the paper:
1. the novelty and the effectiveness of the proposed evaluation method.
2. lack deeper analysis on the combination influence between time and geo-info.

In the below "Questions" part, I will describe them in detail.

**Questions:**

I have two concerns about the paper:
 1. the novelty and the effectiveness of the proposed method.

The first part of the method is img-geo alignment. I found it almost the same as the method in the GEOCLIP paper. I cannot find much difference.  Also in the img-time alignment part, the time encoder has exact the same architecture as the location encoder.

After I finish reading the paper, I think the novelty lies in (1) the temporal metric learning, (2) predict geo and time jointly (3) normalize both month and hours to be continuous. The first one is the most important novelty in my understanding.

1.1 about the temporal metric learning

For the proposed temporal metric learning, I am curious about the training stability of using it,  since the time distribution varies across batches, which will bring instability during training so that calculating KL-loss might be unstable.

Also in Table 2, the cyclic loss did better in month prediction but did worse in hour prediction than L2 loss. In this way,  why the proposed cyclic loss  does not improve the prediction results for hour?  You explain it as "This small difference might be attributed to the fact that the dataset is biased towards daytime ". But L2 loss and cyclic loss experiments used the same dataset. Could you put the hour confusion matrix or prediction error distribution cross time in the supplementary to help readers have a deeper analysis on such phenomenon? Otherwise, the reason you give is not sound, because intuitively, cycle loss is much more reasonable.

1.2 about the prediction results

In Table 1, you compare the time prediction results between time-clip and geotime-clip. There is a sentence " Our experiments demonstrate that GeoTimeCLIP achieves lower errors for month and hour predictions compared to all baselines, without the need for additional metadata." However, GeoTimeCLIP adds geographic location as an additional metadata input compared to TimeCLIP.

Also, the Geo-localization results in Table 5 is not that good. Why GeoTimeCLIP performs worse than GeoCLIP in some metrics for the dataset Im2GPS3k, since  GeoTimeCLIP  uses more input. I cannot find any analysis to such phenomenon. You give the statement for Table 1 that " training a model for both time prediction and geo-localization simultaneously results in richer time representations", so why "training geo and time simultaneously" not results in very good geo-location representation?

2. lack deeper analysis on the combination influence between time and geo-info.

To be honest, I expect more on the joint analysis between time and geo-info, in addition to the retrieval and prediction results. It seems that the downstream task lacks deeper analysis on the combination influence between time and geo-info, since there is a significant correlation between geographical location and time features. For example, in high-latitude regions of the Northern Hemisphere, daylight hours are shorter in winter, and it gets dark earlier. In contrast, in low-latitude regions near the equator, daylight hours remain more stable, and sunset times do not vary significantly. However, I cannot find such kind of analysis.

---

### Official Review · Reviewer_sEKj · 2024-11-09

**Soundness:** 3
**Presentation:** 3
**Contribution:** 1
**Rating:** 3
**Confidence:** 4

**Summary:**

This submission presents a simple method to predict time and geo-location from a given image. This method adopts the CLIP framework which aligns image embedding and text embedding in the representation space by treating time and geo-location as texts. Considering Cyclic nature of time in terms of days and years, text embeddings for time are computed using Random Fourier Features. On the other side, predicting geo-location follows the existing GeoCLIP framework. To evaluate time prediction, this submission also introduce a evaluation metric (i.e., Time Prediction Score). Based on this TPS metric, the proposed method outperforms baselines on the zero-shot time prediction task and yields comparable accuracies on the geo-localization task.

**Strengths:**

1. Yields a good performance on time prediction tasks.
2. Propose a new yet simple time prediction metric.

**Weaknesses:**

1. Lack of novelty

The only novel contributions that the authors can claim in this work are that they propose a task-specific learning framework in the form of CLIP and a loss that takes into account the cyclic nature of time. However, the portion of the CLIP framework that is appropriately designed for the task (i.e., a pair of CLIPs in the unified framework) is very minimal to be considered a crucial contribution. The RFF loss, which was designed to utilize the cyclic nature in terms of frequency, does not seem to make a large contribution. It would be helpful to better understand the proposed loss if the RFF loss values ​​matching the actual cyclic nature can be visualized.

Moreover, since the proposed methods are designed and evaluated for too narrow a task, it is questionable whether they can be used more generally.


2. Insufficient performance

In geo-localization, the performance was worse than some compared methods. If joint training of two tasks in an integrated framework results in better performance on only one task, then the need for joint training must be questionable.

**Questions:**

What is the intuition behind joint training two-CLIP framework? Since two tasks (time prediction and geo-localization) are very different, separate training may be more effective than joint training.

**Details Of Ethics Concerns:**

I don't have any ethics concern.

---

### Meta-Review · Area_Chair_ZmK9 · 2024-12-20

**Metareview:**

In this paper, the authors presented a method (termed GeoTimeCLIP) to estimate both the capture time and geo-location of a given image using a retrieval approach. Specifically, based on the CLIP framework, the visual embedding was aligned with the location and time embeddings, for which the similarity between image and location was modelled by standard contrastive loss and the image-time similarity was using a contrastive loss together with a distance loss. Experiments showed the effectiveness of the proposed method over previous methods. The main strengths of this paper are:
- A new setting of joint embedding was presented and investigated in this paper, which is interesting.
- The newly introduced time prediction metric was favoured by the reviewers.
- The performance of the presented method was shown to be good in the experiments.
- The paper was generally well-written and easy to follow. The related work is also solid.

The main weaknesses of this paper include:
- The novelty of the proposed method. Since the presented method was mainly based on the CLIP framework, and the image-location alignment part is the same as previous work such as GeoCLIP, the image-time part also follows a similar design, reviewers were mainly concerned about the technical novelty.
- The design or the assumption of the technical approach. The authors proposed to jointly align the visual, geo-location, and time in the same embedding space. Such a design is questionable and was not well justified with convincing evidence.
- The problem domain or setting is a bit narrow/niche, making the potential application and interest an issue.
- The experimental analysis was found to be a bit insufficient and lack of depth by the reviewers.

Overall, this paper is well-written and clearly describes the proposed method, the results also show the strength of the method. However, there are a few major concerns not clearly addressed, which need a major revision followed by another round of review.  Specifically, the limited novelty and the questionable design/assumption of the proposed method are the main concerns leading to the decision. Considering the above, the paper in its current form is not ready to be presented at ICLR, but the authors are encouraged to carefully consider all the review comments and revise their paper for future submission.

**Additional Comments On Reviewer Discussion:**

This paper had a good discussion, the authors provided a detailed rebuttal, reviewers checked the responses and most of them were well-engaged with the authors in the authors-reviewers discussion phase. During the AC-reviewers discussion, a further discussion was initiated and some reviewers presented their thoughts. Finally, this paper received 1 Reject, 1 Borderline Reject, and 2 Borderline Accept. Overall, the AC and reviewers agreed on the main strengths and weaknesses of this paper. The final decision was made based on the review comments and the major issues mentioned in the above section.
The AC understand this might be an unpleasant result and the authors may not agree with it, but hope the key points that led to this decision together with the detailed comments from the reviewers could be helpful for the authors in improving their paper.

---

### Decision · Program_Chairs · 2025-01-22

Reject